# Analysis of Priority Conservation Areas Using Habitat Quality Models and MaxEnt Models

**DOI:** 10.3390/ani14111680

**Published:** 2024-06-04

**Authors:** Ahmee Jeong, Minkyung Kim, Sangdon Lee

**Affiliations:** Department of Environmental Science and Engineering, Ewha Womans University, Seoul 03760, Republic of Korea; worldami@ewhain.net (A.J.); enviecol@ewha.ac.kr (M.K.)

**Keywords:** habitat quality, InVEST, MaxEnt, protected areas, gap analysis, endangered species

## Abstract

**Simple Summary:**

This study investigated core habitat areas for yellow-throated martens and leopard cats, two endangered forest species sensitive to habitat fragmentation in Korea. Overlaying the InVEST-HQ and MaxEnt models, priority conservation areas were identified by analyzing gaps in currently protected areas. The core area (14.7%) was mainly distributed in forests such as the Baekdudaegan Mountains Reserve and 12.9% was outside protected areas, and only 1.8% was protected. Conservation priority areas were identified as those with more than 95% forest cover, offering an appropriate habitat for the two species. These findings can be used to identify priority conservation areas through objective habitat analysis and as a basis for protected area designation and assessment of endangered species habitat conservation, thereby contributing to biodiversity and ecosystem conservation.

**Abstract:**

This study investigated core habitat areas for yellow-throated martens (*Martes flavigula*) and leopard cats (*Prionailurus bengalensis*), two endangered forest species sensitive to habitat fragmentation in Korea. Overlaying the InVEST-HQ and MaxEnt models, priority conservation areas were identified by analyzing gaps in currently protected areas. The InVEST-HQ model showed that habitat quality ranged from 0 to 0.86 on a scale from 0 to 1, and the majority of the most suitable areas on the Environmental Conservation Value Assessment Map, designated as grade 1, were derived correctly. The MaxEnt model analysis accurately captured the ecological characteristics of the yellow-throated marten and the leopard cat and identified probable regions of occurrence. We analyzed the most suitable yellow-throated marten and leopard cat habitats by superimposing the two results. Gap analysis determined gaps in existing protected areas and identified priority conservation areas. The core area (14.7%) was mainly distributed in forests such as the Baekdudaegan Mountains Reserve in regions such as Gyeongbuk, Gyeongnam, and Gangwon; 12.9% was outside protected areas, and only 1.8% was protected. The overlap results between protected and non-protected areas were compared with different land use types. Conservation priority areas were identified as those with more than 95% forest cover, offering an appropriate habitat for the two species. These findings can be used to identify priority conservation areas through objective habitat analysis and as a basis for protected area designation and assessment of endangered species habitat conservation, thereby contributing to biodiversity and ecosystem conservation.

## 1. Introduction

Biodiversity is a critical component of Earth’s ecosystems, providing multiple resources and playing an essential role in maintaining the stability and functioning of ecosystems. This biodiversity is intimately connected to our lives and well-being and its conservation is of great value. The loss of biodiversity due to ecosystem degradation also affects the services ecosystems provide humans [1]. The most effective way to conserve ecosystems and biodiversity is to protect habitats and designate areas of high ecological value as protected areas [2]. In Korea, top predators such as tigers (*Panthera tigris altaica*), leopards (*Panthera pardus orientalis*), and wolves (*Canis lupus coreanus*) have become extinct or ecologically irrelevant over the past century, increasing the importance of the remaining predators such as yellow-throated martens (*Martes flavigula*) and medium to large mammals such as leopard cats (*Prionailurus bengalensis*). Yellow-throated martens and leopard cats, both listed as Class II endangered species by the Ministry of Environment, exhibit a wide range of behavior and are highly vulnerable to the effects of habitat fragmentation because of their small population size [3]. Specifically, 63% of South Korea’s territory is covered in forests [4]. Endangered forest-dependent species like the yellow-throated marten and leopard cat are important subjects for studies on forest landscape conservation and the designation of protected areas [5,6]. In addition, yellow-throated martens and leopard cats are flagship species for the Korean ecosystem, and targeting them will increase the likelihood of protecting other populations, which will help conserve biodiversity.

Current domestic wildlife reserves are small in area and have vague designation criteria; there is a need for objective assessment and specific protections for endangered species [4]. Many countries have historically prioritized and managed conservation solely by assessing areas rich in biodiversity, which may underestimate the potential value of species with broader influence ranges. Therefore, numerous recent studies have assessed the diverse functions and values of ecosystems in order to address these limitations [3,7].

Ensembling results by combining multiple models instead of using only a single model has become popular in recent research. This is to overcome the limitations of a single model and produce more reliable results [8,9]. In this study, we followed this approach and combined the results of two or more models. In particular, the InVEST-HQ model has the advantage of being able to consider land use change, but has limitations in its application to endangered species habitats. On the other hand, the MaxEnt model provides more realistic results because it bases its predictions on actual occurrence data of species. Therefore, in this study, we combined the results of these two models, a technique that has been adopted by other studies recently [10,11,12,13].

This study analyzed priority areas for habitat conservation for endangered forest species, such as the yellow-throated marten and leopard cat, that are sensitive to habitat fragmentation. To analyze the habitat of these two species, we (1) evaluated protected areas designated and managed for wildlife habitat protection; (2) habitat quality in Korea was assessed using the InVEST Habitat Quality Model, which evaluates habitat quality in the area using land cover; and (3) used the MaxEnt model to predict the probability of species occurrence, allowing us to identify suitable habitats for endangered species with small populations. Finally, we combined the results of the two models to identify the most suitable core area likely to be used by the yellow-throated marten and leopard cat and conducted a gap analysis with existing protected areas to identify priority conservation areas. The findings provide a basis for habitat conservation for the yellow-throated marten and leopard cat, as well as assist in the designation of new protected areas and evaluation of existing ones.

## 2. Materials and Methods

### 2.1. Study Flow

The overall research flow is depicted in Figure 1. After overlaying the results of the InVEST Habitat Quality and MaxEnt models, we conducted a gap analysis with existing protected areas to identify priority areas for conservation.

### 2.2. Study Species and Site

The yellow-throated marten and leopard cat are classified as least concern by the IUCN Red List of Threatened Species and are listed Class II endangered species by the Ministry of Environment and in the vulnerable category on the National Biodiversity Red List in South Korea. (Figure 2). The yellow-throated marten is found in mountainous areas throughout Asia in dense forests and forested valleys near streams. In contrast, the leopard cat is found throughout Korea in forests and fields, except on Jeju Island and some other islands. While both species have experienced a dramatic decline in their populations in recent years due to deforestation, lack of habitat, and poaching [14], there has been insufficient research on conserving their habitats in Korea. The yellow-throated marten and leopard cat are the flagship species of the Korean ecosystem, and we chose them as targets because of their common use of forests. If we can identify critical habitats through habitat analysis, we can inform the selection of additional protected areas.

The study was conducted across South Korea as follows, and each administrative region was divided into nine regions for area comparison (Figure 3b). South Korea is located between China to the west and Japan to the east (Figure 3a), between 33~43° latitude and 124~132° longitude. It is a peninsula bordered on three sides by the sea and consists of 63% forests. The Baekdudaegan Mountains Reserve stretches from north to south, forming a high plateau in the east and a low one in the west. There are also four distinct seasons, so the vegetation varies with different types of forests depending on the region.

### 2.3. Selection of Protected Areas

This study selected protected areas closely associated with forest wildlife habitats, with yellow-throated marten and leopard cat as the target species: the protected areas designated under the Fourth Basic Plan for Wildlife (2021–2025) and the Baekdudaegan Mountains Reserve, excluding Special Islands that are not habitats for the yellow-throated marten and leopard cat [2]. Finally, a total of five protected areas were extracted from the existing protected areas and built into a 1 km^2^ resolution raster (Figure 3c): the Wildlife Protection Area, the Wildlife Special Protection Area, the Wetland Protected Area, the Nature Park (including National Park, County Park, and Provincial Park), and the Baekdudaegan Mountains Reserve [16].

### 2.4. Analyzing Habitat Quality with the InVEST-HQ Model

#### 2.4.1. InVEST-HQ Model

To analyze the habitat quality for yellow-throated martens and leopard cats, it is crucial to employ multiple metrics and models that can quantitatively assess the diverse benefits of an ecosystem. The InVEST (Integrated Valuation and Environmental Services and Tradeoffs) model used in this study consists of various ecosystem service valuation items that can be valued using relevant variables based on land cover. It is open source and easily accessible, and the analysis results can be visualized on a map [17]. Therefore, it has recently been used as a decision-support model for ecosystem services [13,18].

One of the InVEST models, Habitat Quality (HQ), utilizes land cover to assess an area’s habitat quality. This model can be used as an indicator of biodiversity [17]. The InVEST-HQ model can easily overlap with the species distribution model [13].

#### 2.4.2. Creating Threats and Sensitivity Variables

The threats in the InVEST-HQ model were prioritized based on the threats presented in the ecological papers of the yellow-throated marten and leopard cat, except poaching, which is challenging to incorporate into raster data [19,20,21,22]. Nine threats were selected: residential areas, industrial areas, commercial areas, recreational areas, roads, public utilities, agricultural land (excluding paddy and dry fields), paddy fields, and dry fields. Among these, roads were divided into traffic areas in the land cover map and further classified into three levels based on the maximum speed (MAX_SPD), taken from national standard node-link data (Table 1).

Next, we set the suitability and sensitivity values for the threat factors using the values verified in South Korean research. Habitat suitability and sensitivity can range from 0 to 1, with values closer to 1 indicating higher suitability and sensitivity. Values set by experts in [18] were preferentially used for domestic conditions, and residential values that were not included in their report were derived from other previous studies [23,24]. The proportions of roads and agricultural land were set to match those in previous studies, considering the available information from studies that assessed forested mammals and forests [23,25,26]. All threats except roads (national standard node-link, 2023) were extracted from the land cover map (Environmental Geographic Information Service, 2022). The finalized threat and sensitivity table is presented in Table 1.

This study used the InVEST (v3.14.0) model (https://naturalcapitalproject.stanford.edu/software/invest (accessed on 1 October 2023)) for analysis. In addition, the habitat quality results were compared to the Environmental Conservation Value Assessment Map (ECVAM) for validation, and the habitat excellence areas were examined based on ECVAM ratings [27].

### 2.5. Analyzing Potential Habitats with the MaxEnt Model

#### 2.5.1. MaxEnt Model

There is an increasing need for a species distribution model that can predict the probability of species occurrence in the case of endangered wildlife, where identifying the habitat is challenging due to the small population. The MaxEnt (Maximum Entropy) model is one of the species distribution models that can predict the probability of distribution by utilizing actual species occurrence data and environmental variables [28]. As the data surveyed in Korea only include occurrence data, the MaxEnt model is more suitable than other models [29,30]. For this study, the MaxEnt model was selected to predict potential habitats for the yellow-throated marten and leopard cat.

#### 2.5.2. Creating Occurrence Data and Environmental Variables

The occurrence data of yellow-throated martens and leopard cats were obtained from the 4th National Natural Environment Survey. Spatial autocorrelation (SAC), a measure of the spatial dependence of the data, was determined using the average nearest neighbor index in R (v.4.3.1). Coordinates were clustered to some extent, and to avoid overfitting due to spatial autocorrelation we used the spThin package in R to ensure that each point was at least 1 km^2^ apart and adjusted accordingly. In total, we used 446 occurrence points for the yellow-throated marten and 2379 points for the leopard cat.

The environmental variables used in the MaxEnt model were topography (elevation, gradient, aspect, and topographic wetness), distance (distance from residential areas, used areas, roads, and agricultural land), climate, vegetation (normalized difference vegetation index, diameter, age, and density), and land cover, based on previous studies of yellow-throated martens and leopard cats [6,20,25,31,32,33,34] (Table 2).

All environmental variables were utilized at a spatial resolution of 1 km^2^, and Pearson’s correlation coefficient (*r*) was computed to prevent multicollinearity. Most of the highly correlated variables (|*r*| > 0.7) were climate variables. After removing a proportion of the variables, 18 environmental variables were constructed (Table 2).

For the MaxEnt (v.3.4.4) model (https://biodiversityinformatics.amnh.org/open_source/maxent/ (accessed on 1 October 2023)), we used 70% of the occurrence points as training data for model development and the remaining 30% as test data to verify the model results. The training and test data were selected five times using the cross-validation method. In the model, the background points were set to 10,000, the regularization multipliers were set to 1, and the auto features included linear, quadratic, product, and hinge types. The output from the model was formatted as logistic. This is because the effect on the appearance of a species can be assessed with a value between 0 and 1 through the logistic setting, and an appropriate threshold can be set to generate a binary map marked with suitable and unsuitable habitats [35].

The model’s predictive power was assessed by measuring the AUC (area under cover) of the ROC (receiver operating characteristic) curve. The AUC value of the ROC curve, which can be used to evaluate the accuracy of the MaxEnt model, is close to 1, indicating that the model has a high predictive probability (the explanatory power of the model is considered meaningful when it is around 0.7 or higher) [28]. The jackknife test was also conducted to determine the relative importance of environmental variables in model generation, along with the response curve.

### 2.6. Gap Analysis

After selecting core areas for the yellow-throated martens and leopard cats, we conducted a gap analysis to identify priority conservation areas by comparing the core areas derived from the nested InVEST-HQ and MaxEnt models with existing protected areas. Gap analysis is an analytical method that identifies gaps in the status of different components of wildlife habitat and conservation and determines suitable habitats [7,8].

The habitat quality results of the InVEST-HQ model were used for the gap analysis to extract the habitat quality of yellow-throated marten and leopard cat occurrence points, with the mean value set as the threshold. Areas above the threshold were classified as “excellent” habitat areas, while areas below the threshold were classified as habitat “management” areas [18]. The “maximum test sensitivity plus specificity” threshold from the logistic output of the MaxEnt model was used to identify potential habitat areas. Areas that exceeded the average thresholds for both species were classified as “suitable” areas, while areas below the threshold were classified as “unsuitable” [35].

The converted “excellent” habitat areas and “suitable” areas were overlaid, and the “core area” was selected as habitats expected to be excellent for yellow-throated martens and leopard cats and with a high probability of occurrence. In addition, by overlaying the previously extracted protected areas, we compared the excellent habitat areas, suitable areas, and core habitats outside and inside the protected areas. Finally, we selected priority areas for conservation with Table 3.

## 3. Results and Discussion

### 3.1. Habitat Quality Analysis Results

The results of the habitat quality analysis for yellow-throated martens and leopard cats using the InVEST-HQ model are shown in Figure 4a. Habitat quality ranged from 0 to 0.86 on a scale of 0 (low quality) to 1 (high quality), averaging 0.53 ± 0.27. The redder the color of the map, the higher the habitat quality. We found that the highest habitat quality was centered around significant mountain ranges such as Gangwon and parts of Gyeongbuk, where there is significant forest cover outside of city centers. Areas with high habitat quality have a higher likelihood of biodiversity sustainability [13], and active protection and management of these areas are needed to conserve them.

The average habitat quality value of the yellow-throated marten and leopard cat was calculated to establish the threshold. The final threshold was determined to be 0.67, with 44,021 km^2^ (43.9%) of excellent habitat areas above this value and 56,178 km^2^ (56.1%) of habitat management areas below it (Figure 4d). To evaluate the validity of the habitat quality results, we compared them with the ECVAM [27]. We found that most of the excellent habitat areas for yellow-throated martens and leopard cats were rated as grade 1. This confirms that the excellent habitat areas were appropriately identified based on Korean conditions.

### 3.2. Potential Habitat Analysis Results

In our analysis, the AUC values for the yellow-throated marten and leopard cat were 0.810 and 0.645, respectively (Table 4). In the case of leopard cats, it is difficult to exceed 0.7 on average because the occurrence coordinates are widely distributed across the country, similar to general species, and like previous studies that ran MaxEnt models for leopard cats at the national level, we found that the AUC values were relatively low (0.561, 0.629, and 0.761) [25,32,36]. Therefore, although the AUC value does not indicate a very high accuracy, it can be evaluated as relatively reliable because it aligns with relatively low values in previous studies [25].

In addition, although we have addressed spatial clustering in our study, there is still some degree of SAC remaining. This clustering could be due to differences in survey methods or specific behavioral patterns of the species. Due to time and data constraints, adjustments to the SAC were limited, but future studies will address this issue in more depth with spatial econometric models [37].

The results of the jackknife test to assess the contribution of variables to the distribution of potential habitats for the yellow-throated marten and leopard cat and the relative importance of each variable are shown in Table 4. Of the environmental variables, elevation, slope, and distance from used areas were the most important variables affecting the distribution of yellow-throated martens. Bio7 (temperature annual range), slope, and land cover were the most critical variables for leopard cats. The response curves indicated that yellow-throated martens were more likely to be located at higher elevations, slopes, and distances from used areas, which is similar to the findings of previous studies that have shown that yellow-throated martens are more likely to occur in forests with higher elevations and a more complex canopy structure, especially in broadleaf forests with plant fruits and rodents that can serve as food sources [20,33]. In the case of leopard cats, we found that they were more likely to be located in forests and wetlands with annual temperature differences between 30 °C and 33 °C and low slopes, which aligns with previous studies that showed that leopard cat prefers relatively low elevation, medium-hardwood forests with gentle slopes, and abundant streamside grasslands [19,38]. Therefore, this study’s MaxEnt model of yellow-throated martens and leopard cats supports the results of these previous studies.

The results of predicting potential habitats for yellow-throated martens and leopard cats using the MaxEnt model are shown in Figure 4b,c. The predicted values range from 0 to 1, with redder colors on the map indicating a higher distribution probability. In the case of the yellow-throated marten, the distribution probability was concentrated in areas with high mountain ranges, centered on the Baekdudaegan Mountains Reserve. In the case of leopard cats, the distribution probability was high throughout the country except for some urban centers. Since the leopard cat uses forests as its primary habitat but has ecological characteristics that allow it to live throughout Korea, it is appropriate to establish protected areas centered on major forest areas where leopard cats can live when considering long-term biodiversity conservation [14].

For thresholding, we used 0.42 ± 0.06, the average of the values for both species where the sum of sensitivity and specificity was maximized as the final threshold. The area of possible occurrences with a threshold of 0.42 or suitable areas was 17,943 km^2^ (17.9%), while the area of unsuitable areas with a threshold of 0.42 or lower was 82,256 km^2^ (82.1%) (Figure 4e).

### 3.3. Core Area Analysis Results

To identify the core habitat, we overlaid the habitat results from the InVEST-HQ and MaxEnt analysis, focusing on forests shared by yellow-throated martens and leopard cats (Figure 5). The core area was selected by overlaying the best and most likely habitat areas divided by the thresholds of the InVEST-HQ and MaxEnt models. The core area (red) covered 14,718 km^2^, or 14.7% of the total area; MaxEnt results only (green), 3225 km^2^ (3.2%), InVEST-HQ results only (blue), 29,303 km^2^ (29.2%), and unsuitable habitat areas (gray) covered 52,953 km^2^ (52.9%) (Figure 5).

The difference between the InVEST-HQ and MaxEnt models is visible on the map because the habitat quality value of forests was set to be high to match the ecological characteristics of yellow-throated martens and leopard cats in the input data. The InVEST-HQ model evaluates habitat quality by considering actual land use changes, and it is necessary to refine the habitat quality evaluation system further in conjunction with other models [13]. In this study, MaxEnt results using actual yellow-throated marten and leopard cat occurrence coordinates yielded similar predictions to the ecological characteristics of each species, supporting the utility of MaxEnt models for supporting decision making at the site selection stage in the management plan of protected areas in conjunction with the InVEST-HQ model [11,12].

### 3.4. Priority Areas for Conservation

Finally, we analyzed the gap between protected areas and our analysis-derived core area to identify priority areas for conservation. For this, we overlaid the maps of protected areas on the maps of wildlife habitats and the core area of yellow-throated martens and leopard cats, where they are likely to occur. The overlapping maps identified priority areas for conservation, as shown in Figure 6.

On the conservation priority areas map, the red color scheme indicates core areas in and outside protected areas, the green color scheme indicates suitable areas in and outside protected areas, and the blue color scheme indicates excellent habitat areas in and outside protected areas. Protected areas that do not overlap with the core areas are shown in purple as overprotected areas, and the boundaries of protected areas are also shown in purple. Otherwise, gray indicates unsuitable habitat areas, and yellow indicates the Baekdudaegan Mountains Reserve and prominent forest veins (Figure 6).

The gap analysis sorted administrative regions in order of the largest area of core area outside of the highest priority protected areas for conservation (Figure 7) and identified the proportion of this area in each region (Table 5). The results showed that the highest priority areas for conservation were Gyeongbuk (5932 km^2^), Gyeongnam (2130 km^2^), and Gangwon (1845 km^2^), while Gyeonggi and Jeju had small areas (Figure 6).

The total area of core areas outside protected areas was 12,914 km^2^ (12.9%), while core area inside protected areas was only 1804 km^2^ (1.8%) (Table 5). Thus, we found that most of the core area for yellow-throated martens and leopard cats in this study was not included in protected areas, suggesting that current protected areas could benefit from a reassessment.

The comparison of land use type and overlap inside and outside protected areas showed that more than 95% of the externally predicted area is forested (Figure 8). This indicates that these areas have a high potential for new habitats for conserving the study’s target species, the yellow-throated marten and leopard cat. In particular, core areas in Gyeongbuk (5932 km^2^), Gyeongnam (2130 km^2^), and Gangwon (1845 km^2^), which are currently unprotected, were found to be centered on forest veins such as the Baekdudaegan Mountains Reserve and Nakdong veins. The Baekdudaegan Mountains Reserve is an important migration corridor and habitat for critical plants and animals, including endangered species [4]. New habitat candidates centered on forests should be considered for designation as protected areas, reflecting the ecological characteristics of yellow-throated martens and leopard cats, which use forests as their primary habitat.

## 4. Conclusions

This study focused on the yellow-throated marten and leopard cat—two endangered forest species that are sensitive to habitat fragmentation—by superimposing the results of the InVEST-HQ and MaxEnt models to analyze core areas with high habitat quality and high occurrence potential. A gap analysis was conducted to identify gaps in existing protected areas and priority areas for conservation.

The “core area” of the yellow-throated marten and leopard cat, which is the overlap between excellent areas and suitable areas, was analyzed to be 14,718 km^2^, or 14.7% of the national area. Of this, “core area outside protected areas”, which does not overlap with protected areas, was 12,914 km^2^ (12.9%), and “core area inside protected areas” was only 1804 km^2^ (1.8%). This suggests that much of the core habitat is not included in protected areas and that protected areas need to be adequately assessed.

These findings suggest that among the “core area outside protected areas”, the highest priority areas for conservation are located in areas such as Gyeongbuk (5932 km^2^), Gyeongnam (2130 km^2^), and Gangwon (1845 km^2^) and are centered on forests, especially the Baekdudaegan Mountains Reserve and Nakdong veins. As the area of forests that serve as the primary habitat for wildlife in Korea continues to decline, a comprehensive ecosystem conservation plan is needed to protect existing forests and designate new protected areas. In particular, when comparing the overlap of land use types and protected areas in this study, areas with more than 95% forest cover were identified as forest areas, appropriately designated as yellow-throated marten and leopard cat protected areas.

The results of this study provide an objective habitat analysis of yellow-throated martens and leopard cats to identify priority areas for conservation, which can be used as a basis for the evaluation and designation of protected areas to conserve habitats for threatened species for biodiversity and ecosystem conservation. Future research should consider various additional taxonomic groups to inform the selection of more effective protected areas. Additionally, it is necessary to further develop the habitat assessment method by supplementing the inputs of the InVEST-HQ and MaxEnt models and comparing them with other ecosystem service models or species distribution models.

## Figures and Tables

**Figure 1 animals-14-01680-f001:**
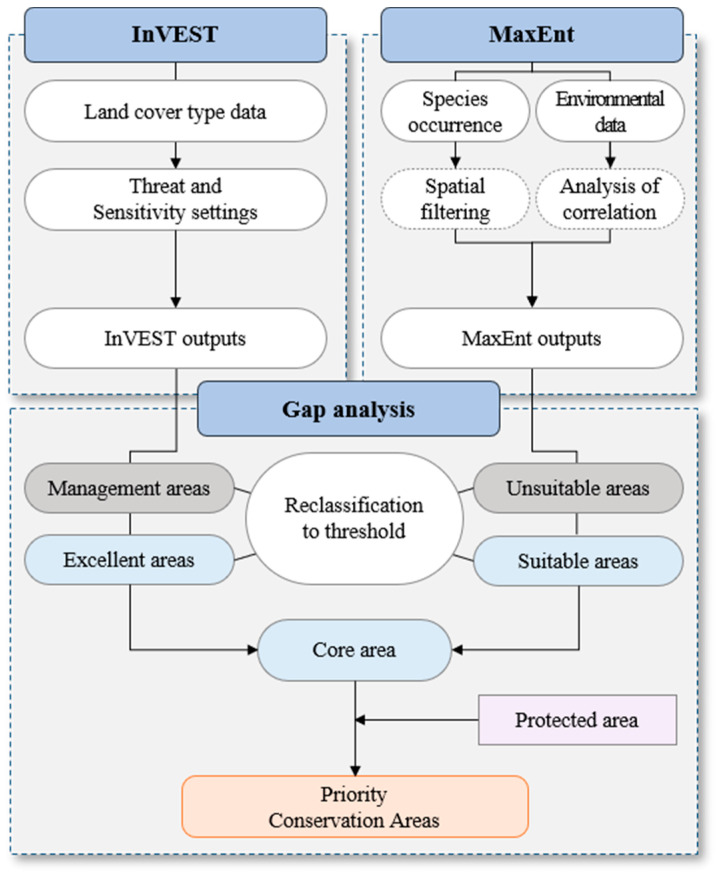
Study flow.

**Figure 2 animals-14-01680-f002:**
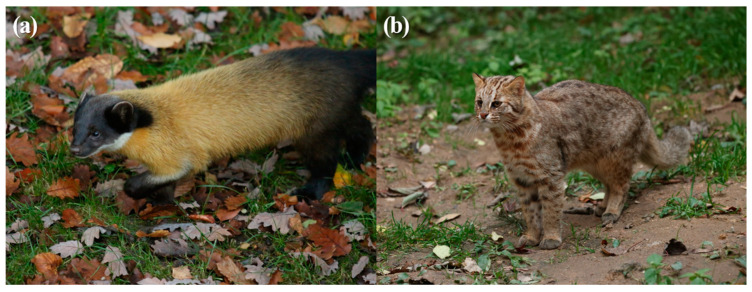
Study species: (**a**) yellow-throated marten (*Martes flavigula*), and (**b**) leopard cat (*Prionailurus bengalensis*) [15].

**Figure 3 animals-14-01680-f003:**
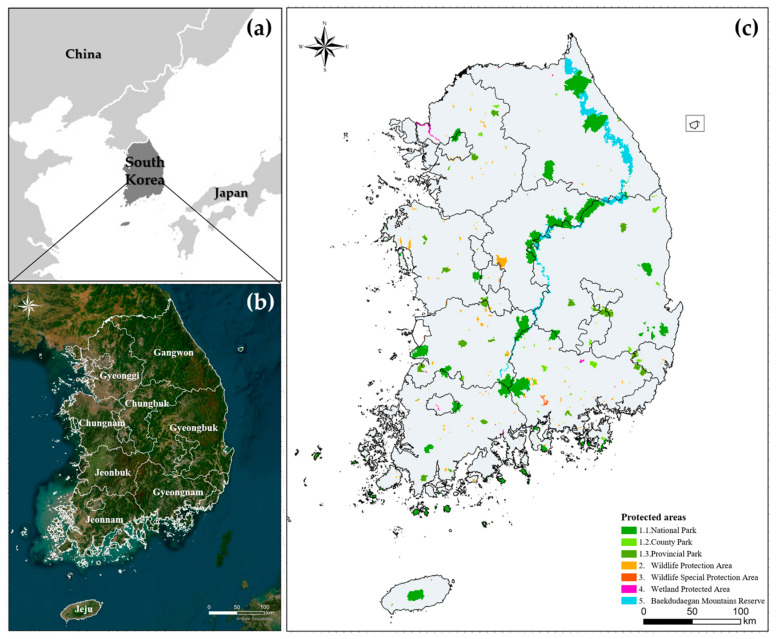
(**a**) Study site; (**b**) provinces in Republic of Korea; and (**c**) protected areas selected in the study.

**Figure 4 animals-14-01680-f004:**
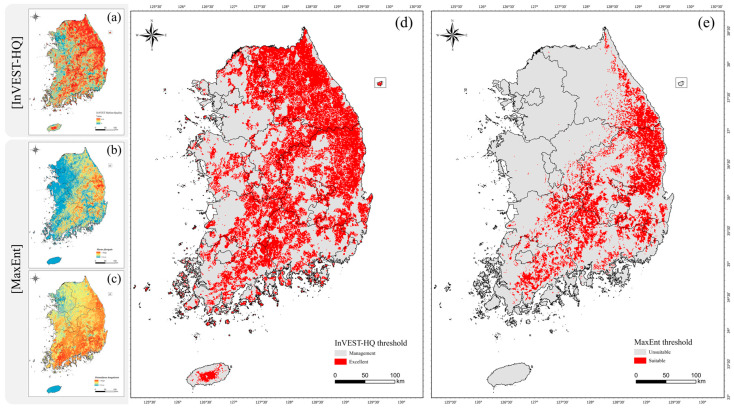
(**a**) InVEST habitat quality map; (**b**) potential distribution map of the yellow-throated marten; (**c**) potential distribution map of the leopard cat; (**d**) excellent/management areas of habitat quality map; and (**e**) suitable/unsuitable areas of potential distribution map.

**Figure 5 animals-14-01680-f005:**
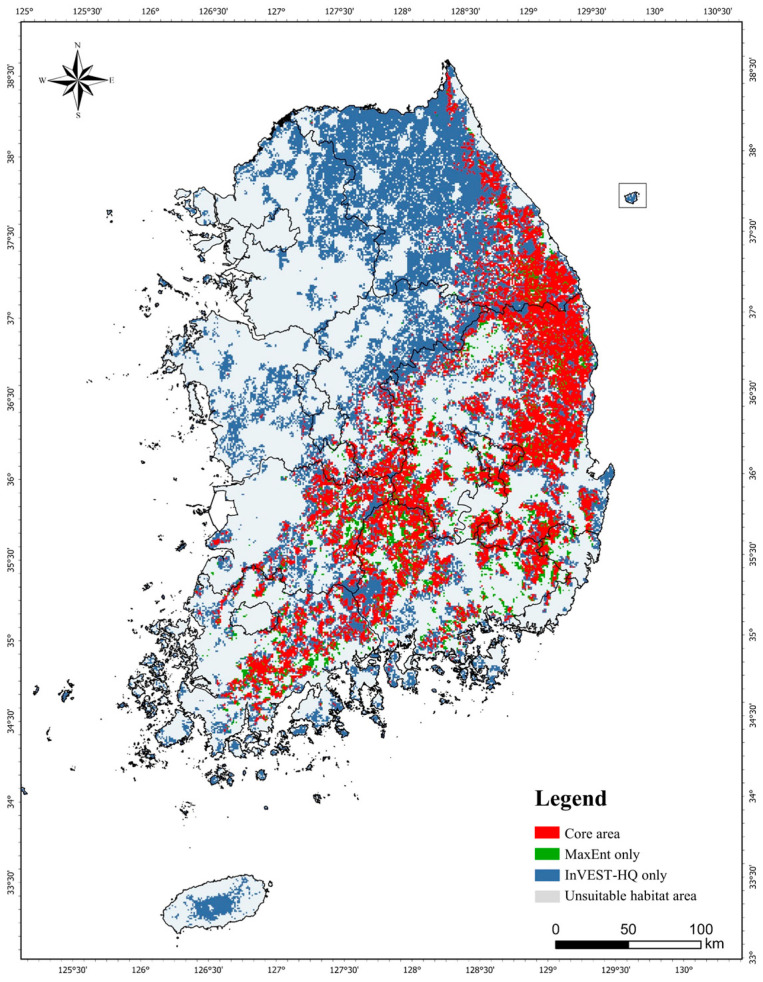
Mapping overlay of the core area map by MaxEnt and InVEST-HQ.

**Figure 6 animals-14-01680-f006:**
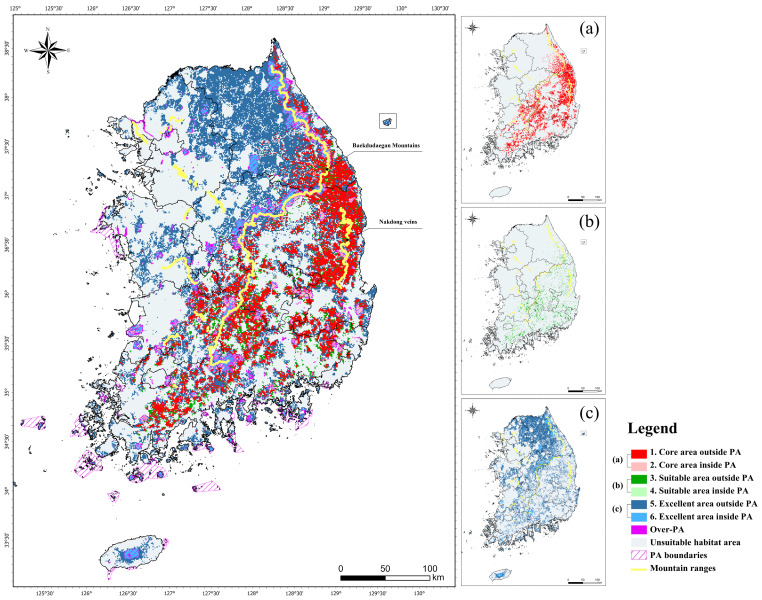
Gap analysis results for conservation priority area: (**a**) core area outside/inside PA, (**b**) suitable area outside/inside PA, and (**c**) excellent area outside/inside PA.

**Figure 7 animals-14-01680-f007:**
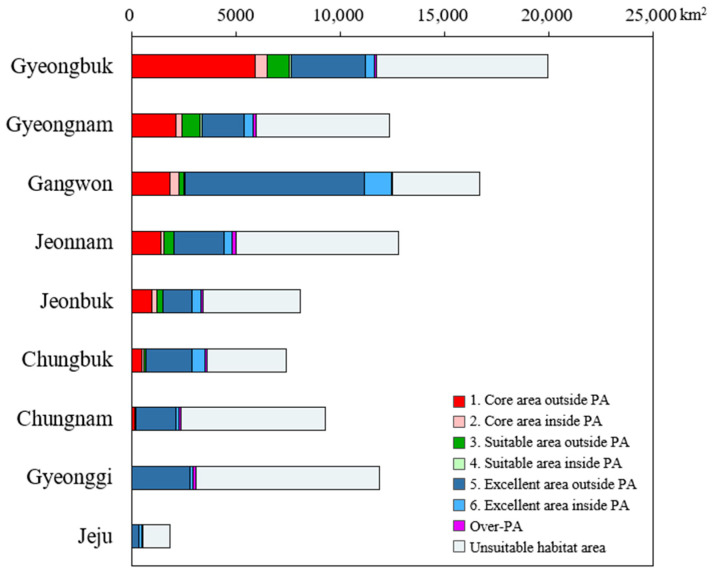
Mapping overlay of core area map by MaxEnt and InVEST-HQ.

**Figure 8 animals-14-01680-f008:**
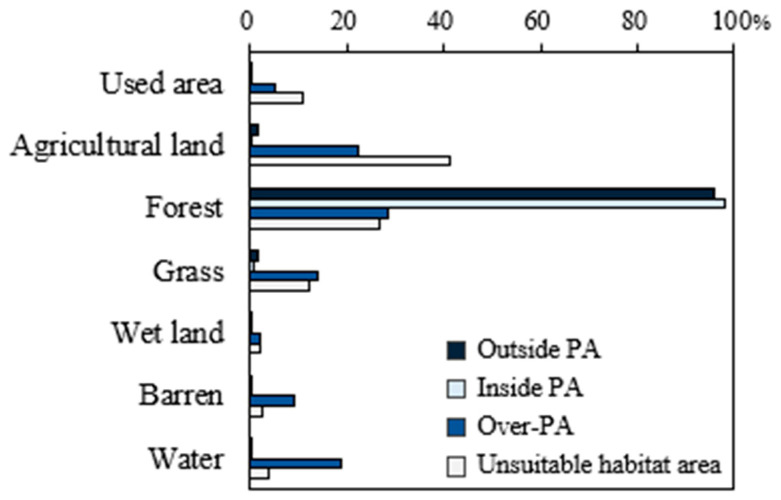
Land use type for gap analysis result areas.

**Table 1 animals-14-01680-t001:** Threat and sensitivity values for the InVEST Habitat Quality model input for yellow-throated marten and leopard cat.

**Threat**	**Description**	**HS ^1^**	**Sensitivity to Threat ^2^**										
			**Res**	**Ind**	**Com**	**Rec**	**Road 1**	**Road 2**	**Road 3**	**Pub**	**Agri**	**Pad**	**Dry**
	**Weight**	-	0.58	0.88	0.88	0.88	0.59	0.40	0.20	0.88	0.57	0.67	0.57
	**Maximum Distance**	-	3.8	5.9	5.9	5.9	2.4	2.0	1.6	5.9	3.4	4.7	4.7
	**Decay ^3^**	-	Expo	Line	Line	Line	Expo	Expo	Expo	Line	Line	Line	Line
**Sensitivity**	Residential area	0.05	0	0	0	0	0	0	0	0	0	0	0
	Industrial area	0.05	0	0	0	0	0	0	0	0	0	0	0
	Commercial area	0.05	0	0	0	0	0	0	0	0	0	0	0
	Recreational area	0.05	0	0	0	0	0	0	0	0	0	0	0
	Road	0.05	0	0	0	0	0	0	0	0	0	0	0
	Public utility area	0.05	0	0	0	0	0	0	0	0	0	0	0
	Paddy field	0.25	0.33	0.41	0.41	0.41	0.25	0.15	0.05	0.41	0.31	0	0.10
	Dry field	0.30	0.33	0.41	0.41	0.41	0.25	0.15	0.05	0.41	0.31	0.16	0
	Facility plantation	0.25	0.33	0.41	0.41	0.41	0.25	0.15	0.05	0.41	0.31	0.16	0
	Orchard	0.30	0.33	0.41	0.41	0.41	0.25	0.15	0.05	0.41	0.31	0.16	0
	Other plantations	0.40	0.33	0.41	0.41	0.41	0.25	0.15	0.05	0.41	0.31	0.16	0
	Broadleaved forest	0.86	0.60	0.75	0.75	0.75	0.52	0.42	0.32	0.75	0.66	0.61	0.51
	Coniferous forest	0.86	0.60	0.75	0.75	0.75	0.52	0.42	0.32	0.75	0.66	0.61	0.51
	Mixed forest	0.86	0.60	0.75	0.75	0.75	0.52	0.42	0.32	0.75	0.66	0.61	0.51
	Natural grassland	0.50	0.36	0.45	0.45	0.45	0.33	0.23	0.13	0.45	0.46	0.41	0.41
	Artificial grassland	0.34	0.36	0.45	0.45	0.45	0.33	0.23	0.13	0.45	0.46	0.41	0.41
	Inland wetland	0.70	0.56	0.70	0.70	0.70	0.55	0.45	0.35	0.70	0.75	0.85	0.75
	Coastal wetland	0.70	0.56	0.70	0.70	0.70	0.55	0.45	0.35	0.70	0.75	0.85	0.75
	Bare ground	0.08	0.11	0.14	0.14	0.14	0.05	0.04	0.03	0.14	0.15	0.15	0.15
	Artificial ground	0.08	0.11	0.14	0.14	0.14	0.05	0.04	0.03	0.14	0.15	0.15	0.15
	Inland water	0.65	0.58	0.73	0.73	0.73	0.55	0.45	0.35	0.73	0.65	0.75	0.65
	Marine water	0.65	0.58	0.73	0.73	0.73	0.55	0.45	0.35	0.73	0.65	0.75	0.65

^1^ HS: habitat suitability; ^2^ Res: residential area, Ind: industrial area, Com: commercial area, Rec: recreational area, Road 1: ~60 km/h, Road 2: 60~80 km/h, Road 3: 80 km~/h, Agri: agricultural land, Pub: public utility, Pad: paddy field, Dry: dry field; ^3^ Decay: Expo: exponential, Line: linear.

**Table 2 animals-14-01680-t002:** Environmental variables used for MaxEnt modeling.

Classification	Code	Variables	Type	Source
Topography	DEM	Elevation	Continuous	Digital Elevation Model (National Geographic Information Institute, 2014)
	Slope	Gradient		
	Aspect	Aspect		
	TWI	Topographic Wetness Index		Topographic Wetness Index (Korea Institute of Geoscience and Mineral Resources, 2019)
Distance	Res	Distance from residential area		Land cover map (Environmental Geographic Information Service, 2022)
	Used	Distance from used area		
	Road	Distance from road		
	Agri	Distance from agricultural land		
	Water	Distance from water		
Climate	Bio3	Isothermality (Bio2/Bio7) (×100)		WorldClim (1970~2000)
	Bio7	Temperature annual range (Bio5-Bio6)		
	Bio12	Annual precipitation		
	Bio13	Precipitation of wettest month		
Vegetation	NDVI	Normalized Difference Vegetation Index		Normalized Difference Vegetation Index (Korea Institute of Geoscience and Mineral Resources, 2022)
	DMCLS	Diameter of forest	Categorical	1: 5000 Forest type map (Korea Forest Service, 2022)
	AGCLS	Age of forest		
	DNST	Density of forest		
Land cover	LULC	Land cover type		Land cover map (Environmental Geographic Information Service, 2022)

**Table 3 animals-14-01680-t003:** Determining the ranking of conservation priorities and explaining the overlay map.

Ranking	Overlay Map			Explain
	InVEST-HQ	MaxEnt	PA ^1^	
1	O	O		Core area outside PA
2	O	O	O	Core area inside PA
3		O		Suitable area outside PA
4		O	O	Suitable area inside PA
5	O			Excellent area outside PA
6	O		O	Excellent area inside PA
-			O	Over-PA
-				Unsuitable habitat area

^1^ PA: protected area.

**Table 4 animals-14-01680-t004:** AUC values, relative importance of environmental variables, and jackknife test by MaxEnt model.

Species	AUC	Percent Contribution	Jackknife
*Martes flavigula*	0.810	**DEM** ^1^ > Bio3 > Bio7 > **Used** > **Slope**	**DEM** > **Slope** > **Used** > DMCLS > AGCLS
*Prionailurus bengalensis*	0.645	**Bio7** > **Slope** > DEM > **LULC** > Used	**Bio7** > Bio3 > **Slope** > Bio12 > **LULC**

^1^ Bold variables indicate those common to both percent contribution and jackknife tests.

**Table 5 animals-14-01680-t005:** Percentage and conservation priority ranking area (km^2^) by administrative district.

Ranking	1	2	3	4	5	6	-	-
Map Color (Figure 6 and Figure 7)	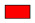	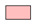	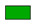	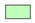	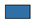	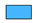	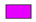	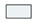
District	Core AreaOutside PA	Core AreaInside PA	Suitable Area Outside PA	Suitable AreaInside PA	Excellent AreaOutside PA	Excellent AreaInside PA	Over-PA	Unsuitable HabitatArea
Gyeongbuk	19,914	5932(29.79%)	570(2.86%)	1054(5.29%)	72(0.36%)	3559(17.87%)	432(2.17%)	92(0.46%)	8203(41.19%)
Gyeongnam	12,373	2130(17.21%)	293(2.37%)	877(7.09%)	62(0.50%)	2051(16.58%)	399(3.22%)	140(1.13%)	6421(51.90%)
Gangwon	16,664	1845(11.07%)	425(2.55%)	239(1.43%)	25(0.15%)	8640(51.85%)	1259(7.56%)	82(0.49%)	4149(24.90%)
Jeonnam	12,790	1407(11.00%)	143(1.12%)	463(3.62%)	21(0.16%)	2415(18.88%)	368(2.88%)	195(1.52%)	7778(60.81%)
Jeonbuk	8087	952(11.77%)	260(3.22%)	269(3.33%)	25(0.31%)	1408(17.41%)	398(4.92%)	107(1.32%)	4668(57.72%)
Chungbuk	7397	505(6.83%)	86(1.16%)	83(1.12%)	2(0.03%)	2238(30.26%)	614(8.30%)	80(1.08%)	3789(51.22%)
Chungnam	9263	142(1.53%)	27(0.29%)	32(0.35%)	1(0.01%)	1936(20.90%)	126(1.36%)	102(1.10%)	6897(74.46%)
Gyeonggi	11,861	1(0.01%)	0(0.00%)	0(0.00%)	0(0.00%)	2801(23.62%)	156(1.32%)	122(1.03%)	8781(74.03%)
Jeju	1850	0(0.00%)	0(0.00%)	0(0.00%)	0(0.00%)	356(19.24%)	147(7.95%)	16(0.86%)	1331(71.95%)
Total	100,199	12,914(12.89%)	1804(1.80%)	3017(3.01%)	208(0.21%)	25,404(25.35%)	3899(3.89%)	936(0.93%)	52,017(51.91%)

## Data Availability

The data that support the findings of this study are available from the corresponding author, upon reasonable request.

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
