# Peer review of "Analysis of Priority Conservation Areas Using Habitat Quality Models and MaxEnt Models"

_animals, 2024, doi:10.3390/ani14111680_

Round 1
Reviewer 1 Report
Comments and Suggestions for Authors
Animals-2996282
General comments
This paper uses two ecological models to assess the best locations in South Korea to prioritize for novel protected areas for yellow-throated martens and leopard cats. Conservation of these two carnivores will benefit from recommendations made by this paper and it is certainly worth publication after some minor improvements are made, namely to explanations for the analytical approach.
It is not clear why the two different models were used in this paper. Either MaxEnt or the InVEST model appear to be able to incorporate all of the variables used into a single model which would integrate the effects of both habitat quality and threat variables. Whereas these two models may provide different conclusions, it is important to clearly define the rationale for using both.
Inline comments
31 to which survival is this sentence referring? Grammar
43 Make a new sentence
51 The diverse functions and values that may come from conserving yellow-throated martens and leopard cats is not identified in this introduction. More background on how these two species may provide broader ecological value should be provided.
5. 6. The InVEST model may be obscure enough that most readers will not be familiar with it and a brief introduction to how it works would be useful here
102. Is not clear in this sentence if the authors are referring to existing protected areas or new protected areas identified in their study. could use clarification
136, the South Korean research referred to here need to be referenced
16 2. Which package in R?
164 this section would benefit from a description of how often these two species were not observed at survey points in the fourth national natural environment survey
185 more details are needed to understand how the core areas were selected. Were they selected using the max ant model or with the InVEST model?
198 was there any consideration of areas already developed or used by humans that were not considered suitable for protected areas
Author Response
Dear Reviewer #1,
Thank you for your thorough review and constructive feedback on our manuscript. We value your insights and have taken great care to address the concerns and suggestions you raised. Your comments have been instrumental in enhancing the quality and clarity of our research. Here is a detailed response to each of the points you have noted:
|
Comments |
Response to comments |
|
It is not clear why the two different models were used in this paper. Either MaxEnt or the InVEST model appear to be able to incorporate all of the variables used into a single model which would integrate the effects of both habitat quality and threat variables. Whereas these two models may provide different conclusions, it is important to clearly define the rationale for using both. |
We made the following changes: (Lines 57~66) Ensembling results by combining multiple models instead of using only a single model has become popular in recent research. This is to overcome the limitations of a single model and produce more reliable results (Araújo & New., 2007; Marmion et al., 2009). In this study, we followed this approach and combined the results of two or more models. In particular, the InVEST-HQ model has the advantage of being able to consider land use change, but has limitations in its application to endangered species habitats. On the other hand, the MaxEnt model provides more realistic results because it bases its predictions on actual occurrence data of species. Therefore, in this study, we combined the results of these two models, a technique that has been adopted by other studies recently (Menon et al., 2001; Seto et al., 2012; Huang et al., 2022). |
|
(31) to which survival is this sentence referring? Grammar |
We made the following changes: (Line 31) Well-being |
|
(43) Make a new sentence |
We made the following changes: (Lines 43~47) Specifically, 63% of South Korea's territory is covered in forests (ME, 2018). Endangered forest-dependent species like the yellow-throated marten and leopard cat are important subjects for studies on forest landscape conservation and the designation of protected areas (Mergey et al., 2011; Lee et al., 2017). |
|
(51) The diverse functions and values that may come from conserving yellow-throated martens and leopard cats is not identified in this introduction. More background on how these two species may provide broader ecological value should be provided. |
We made the following changes: (Lines 47~49) In addition, yellow-throated martens and leopard cats are flagship species for the Korean ecosystem, and targeting them will increase the likelihood of protecting other populations, which will help conserve biodiversity. |
|
(56) The InVEST model may be obscure enough that most readers will not be familiar with it and a brief introduction to how it works would be useful here |
We made the following changes: (Lines 71~73) habitat quality in Korea was assessed using the InVEST Habitat Quality Model, which evaluates habitat quality in the area using land cover data. |
|
(102) Is not clear in this sentence if the authors are referring to existing protected areas or new protected areas identified in their study. could use clarification |
We made the following changes: (Lines 120~122) Finally, a total of five protected areas were extracted from the existing protected areas and built into a 1km2 resolution raster. |
|
(136) the South Korean research referred to here need to be referenced |
Please see “KEI (Korea Environment Institute). Development of Decision Supporting Framework to Enhance Natural Capital Sustainability: Focusing on Ecosystem Service Analysis. Korea Environment Institute. 2015, 3479-3651.” in the reference list. |
|
(162) Which package in R? |
We made the following changes: (Line 187) we used the spThin package in R to ensure that each point was at least 1km2 apart and adjusted accordingly |
|
(164) this section would benefit from a description of how often these two species were not observed at survey points in the fourth national natural environment survey |
Thank you for your suggestion to include data on the frequency at which these two species were not observed during the 4th national natural environment survey. However, our current dataset only includes observed occurrences and lacks systematic data on non-observations. Due to this limitation, we are unable to provide a comprehensive analysis of the non-observation frequencies for these species. We recognize the importance of this aspect and suggest that future surveys include both presence and absence data to enable a more detailed ecological analysis and improve the model's accuracy. |
|
(185) more details are needed to understand how the core areas were selected. Were they selected using the max ant model or with the InVEST model? |
We made the following changes: (Lines 222~224) After selecting core areas for the yellow-throated martens and leopard cats, we conducted a gap analysis to identify priority conservation areas by comparing the core areas derived from the nested InVEST-HQ and MaxEnt models with existing protected areas. |
|
(198) was there any consideration of areas already developed or used by humans that were not considered suitable for protected areas |
Negative, the study did not consider areas that are already developed or directly used by humans. The main focus of the study was to explore the potential for protected areas in areas with a well-preserved natural state. Therefore, we first predicted suitable habitat for the study species and analyzed the gap between protected areas and gaps. |
Reviewer 2 Report
Comments and Suggestions for Authors Dear Authors,This manuscript addresses a very interesting and important topic.
Unfortunately, it does not have a very well thought out structure. I also have many questions about modelling. I recommend you
to work out the modelling part properly and redo (if necessary) the results. In its present form, the manuscript cannot be accepted to the Animals
journal.
2.1. Study flow
I would like to see a more detailed diagram with specific steps and actions.
The name of figures & tables should be self-explanatory and self-sufficient,
i.e. understandable without referring to the main text. This applies to the
entire text!
2.2. Study species and site
73-85. This belongs in the Introduction chapter. It should be moved there.
Since it is about the choice of research objects and not about the material
as such.
89-90. This is completely insufficient to characterize the study area. A full
description should be given, including at least topography, climate, and main
vegetation types.
94-97. There is no point in explaining what protected areas are in this chapter.
This information would also be better moved to the Introduction.
All 5 PAs should be indicated on the map in Figure 1.
113. Please provide a reference to the model and its authors. Base article,
website, etc. If it has open access, there should be a website link.
117-118. Please provide a reference/link.
118-120. You have already written about it in the Introduction. You can only
keep «InVEST-HQ model can easily overlap with the species distribution model».
And provide a reference to this statement.
Table 1. The name of tables should be self-explanatory and self-sufficient,
i.e. understandable without referring to the main text.
141-142. Where did you get the landcover map and road data from?
You should provide a reference in the text.
144. Please provide a link to the InVEST software.
150-158. This is trivial and general information. There is no point in
writing about it. Especially not in this chapter. But you have written
absolutely nothing about the MaxEnt model building themselves. For example:
What number of background points did you take? How did you select them?
Did you use any mask layer? Why didn't you use bias-file? Which area was used as the training area, and which one as the projection area? It's not at all clear from the manuscript.
What was the basic purpose of modeling, what hyperparameters were used,
how they were selected (model tuning), what feature types (and why) were used,
what output type was used? You cannot select the default settings in MaxEnt
for choosing hyperparameters (feature types and regularization multiplier).
These settings should be determined using special algorithms (data-driven or
model-driven). For an example, please see Muscarella et al. (2014) or the
genetic algorithm (Vignali et al., 2020).
It's better to use spatial block cross validation to achieve independent test and training datasets if you don't have other data to test (please, see Valavi et al., 2019). In your work, you didn't take into account spatial autocorrelation (SAC) in any way, except for partly solving the problem of multicollinearity in variables.
All these are mandatory parameters and model settings that should be written
about in the article. You mentioned about logistic output only on line 195.
Why not cloglog instead? For more details please see Sillero & Barbosa (2021)
and Zurell et al. (2020).
160-162. Where is the evidence that you don't have spatial autocorrelation
(SAC)? Please note that SAC and spatial clustering are different things.
You are writing about spatial clustering. It's very easy to check. For example,
Average Nearest Neighbor Index (ANNI) in ArcGIS or R. SAC is very difficult
to check, because MaxEnt does not output model residuals. For more details please see Sillero & Barbosa (2021).
162. Where is the reference on R software?
171. Table 2. How did you calculate the distances? Using Euclidean distances?
Earlier you wrote that you have mountains on your study area. Since you have not
described your study area, it is difficult to understand what kind of mountains they are (what altitude range). In any case, it is wrong to calculate distances for mountains in a straight line (Euclidean) without taking altitude into account. You have to take the altitude into account for this (for example
Path Distance).
For more details please see Sillero & Barbosa (2021) and
Pshegusov & Chadaeva (2024).
173-176. Why did you use Pearson instead of Spearman? Isn't that a
parametric test? Do you have a normal distribution of variables?
Why didn't you do a VIF test? 18 variables is too much and lead to model
overfitting. The recommended number is a maximum of 15.
177. Where is the reference on Maxent software? As I understood you used
GUI for this. Today, working with MaxEnt in the graphical user interface (GUI)
of Java application is not very convenient. The GUI is very limited in settings
(model tuning) and quality estimation (model testing or evaluation). Model
building should be done entirely in R environment. For example, you can use
the sdm, dismo, maxnet, kuenm, or biomod2 packages, but I especially
recommend you the relatively new SDMtune package (please, see
Vignali et al., 2020).
177-178. Dependent quality estimates (data splitting) will not be able to give
reliable predictions.
179-181. AUC is a very weak quality criterion for MaxEnt. AUC has been
heavily criticized and is not recommended for use as a standard in model
quality estimation (Please, see Lobo et al., 2008; Veloz, 2009;
Warren & Seifert, 2011).You should always give multiple quality metrics.
For example, continuous Boyce index (CBI; please see Hirzel et al., 2006),
AUCtest, AUCdiff or True Skill Statistic (TSS). For more details please see
Sillero & Barbosa (2021).
I don't really understand how the overall suitable/unsuitable habitat map
(Fig. 4e) came out for both species at once? Sorry but I did not find
information about this in the Materials and Methods (M&M) chapter.
Please explain more clearly.
229-231. These are not results. Please place this in M&M section.
238-239. This is a very controversial statement because AUC is very
dependent on the spatial clustering of occurrence points. Until you show that
it does not exist, it is too early to talk about meaning of AUC values.
And of course, it is highly desirable to publish the ODMAP protocol if you are
writing an article on SDM. Please, see Zurell et al. (2020) for this.
These are just some of the comments. I recommend that you definitely read
the following papers and take into account their recommendations.
- Sillero, N., Barbosa, A.M. 2021. Common mistakes in ecological niche models // International Journal of Geographical Information Science. V. 35. â„–. 2. P. 213-226. https://doi.org/10.1080/13658816.2020.1798968
- Merow, C.; Smith, M.J.; Silander, J.A., Jr. A practical guide to MaxEnt for
modeling species’ distributions: What it does; and why inputs and settings
matter. Ecography 2013, 36, 1058–1069.
- Phillips, S.J.; Anderson, R.P.; Dudík, M.; Schapire, R.E.; Blair, M.E. Opening
the black-box: An open-source release of Maxent. Ecography 2017, 40,
887–893.
- Araújo, M.; Anderson, R.; Barbosa, A.M.; Beale, C.; Dormann, C.; Early, R.;
Garcia, R.; Guisan, A.; Maiorano, L.; Naimi, B.; et al. Standards for distribution
models in biodiversity assessments. Sci. Adv. 2019, 5, eaat4858.
- Zurell, D.: Franklin, J.; König, C.; Bouchet, P.J.; Dormann, C.F.; Elith, J.;
Fandos, G.; Feng, X.; Guillera-Arroita, G.; Guisan, A.; et al. A standard
protocol for reporting species distribution models. Ecography 2020, 43,
1261–1277.
- Pshegusov R.H., Chadaeva V.A. 2024. Distribution modelling of the
Caucasian endemic Fritillaria latifolia against the background of climate
change // Nature Conservation Research. Vol. 9(1). P. 45–57.
https://dx.doi.org/10.24189/ncr.2024.005
- Muscarella, R. ENMeval: An R package for conducting spatially independent
evaluations and estimating optimal model complexity for MaxEnt ecological
niche models / R. Muscarella, P. J. Galante, M. Soley-Guardia, R. A. Boria,
J. M. Kass, M. Uriarte, R. P. Anderson // Methods in Ecology and Evolution.
2014. V. 5. P. 1198-1205. https://doi.org/10.1111/2041-210X.12261
- Lobo, J. M.; Jiménez-Valverde, A.; Real, R. AUC: a misleading measure of
the performance of predictive distribution models // Global Ecology and
Biogeography. – 2008. – V.17. – P. 145-151.
https://doi.org/10.1111/j.1466-8238.2007.00358.x
- Veloz, S. D. Spatially autocorrelated sampling falsely inflates measures of
accuracy for presence-only niche models // Journal of Biogeography. 2009.
V. 36. P. 2290-2299. https://doi.org/10.1111/j.1365-2699.2009.02174.x
- Warren, D. L.; Seifert, S.N. Ecological niche modeling in Maxent: the
importance of model complexity and the performance of model selection
criteria // Ecological Applications. – 2011. – V. 21. – P. 335-342.
https://doi.org/10.1890/10-1171.1
- Varela, S. Environmental filters reduce the effects of sampling bias and
improve predictions of ecological niche models / S. Varela, R. P. Anderson, R. García-Valdés, F. Fernández-González // Ecography. – 2014. – V. 37. – P. 001-008. https://doi.org/10.1111/j.1600-0587.2013.00441.x
- Valavi, R. blockCV: An R package for generating spatially or environmentally
separated folds for k-fold cross-validation of species distribution models
/ R. Valavi, J. Elith, J. J. Lahoz-Monfort, G. Guillera-Arroita // Methods in
Ecology and Evolution. – 2019. – V. 10. – P. 225-232. https://doi.org/10.1111/2041-210X.13107
- Vignali, S. SDMtune: An R package to tune and evaluate species distribution
models / S. Vignali, A. G. Barras, R. Arlettaz, V. Braunisch // Ecology and
Evolution. – 2020. – V. 10. – â„–. 20. – P. 11488-11506.
https://doi.org/10.1002/ece3.6786
- Hirzel, A.H.; Le Lay, G.; Helfer, V.; Randin, C.; Guisan, A. Evaluating the
ability of habitat suitability models to predict species presences. Ecol. Model.
2006, 199, 142–152.
Author Response
Dear Reviewer #2,
Thank you for your thorough review and constructive feedback on our manuscript. We value your insights and have taken great care to address the concerns and suggestions you raised. Your comments have been instrumental in enhancing the quality and clarity of our research. Here is a detailed response to each of the points you have noted:
|
Comments |
Answer |
|
(2.1. Study flow) I would like to see a more detailed diagram with specific steps and actions. The name of figures & tables should be self-explanatory and self-sufficient, i.e. understandable without referring to the main text. This applies to the entire text! |
(Figure 1) We've revised the Tables and Figures to be more detailed. |
|
(73-85) This belongs in the Introduction chapter. It should be moved there. Since it is about the choice of research objects and not about the material as such. |
(Lines 89~101) To make the description of the species more specific, they are treated as Materials and are included in this chapter to describe them along with the study area where they are found. |
|
(89-90) This is completely insufficient to characterize the study area. A full description should be given, including at least topography, climate, and main vegetation types. |
We made the following changes: (Lines 106~111) South Korea is located between China to the west and Japan to the east, between 33~43° latitude and 124~132° longitude. It is a peninsula bordered on three sides by the sea and consists of 63% forests. The Baekdudaegan Mountains Reserve stretches from north to south, forming a high plateau in the east and a low one in the west. There are also four distinct seasons, so the vegetation varies with different types of forests depending on the region. |
|
(94-97) There is no point in explaining what protected areas are in this chapter. This information would also be better moved to the Introduction. |
The same thing was in the introduction, so I removed it. |
|
All 5 PAs should be indicated on the map in Figure 1. |
(Figure 3-c) Added an illustration of a protected area to figure 3. |
|
(113) Please provide a reference to the model and its authors. Base article, website, etc. If it has open access, there should be a website link. |
Added the reference “Natural Capital Project, 2023” is located at line 136 |
|
(117-118) Please provide a reference/link. |
Added the reference “Natural Capital Project, 2023” and the link is located at line 136, 166. Once you download the InVEST model, you can choose which model you want to use when running the model. |
|
(118-120) You have already written about it in the Introduction. You can only keep «InVEST-HQ model can easily overlap with the species distribution model». And provide a reference to this statement. |
We made the following changes: (Line 137) InVEST-HQ model can easily overlap with the species distribution model (Huang et al., 2022). |
|
(Table 1) The name of tables should be self-explanatory and self-sufficient, i.e. understandable without referring to the main text. |
We made the following changes: (Table 1) Threat and sensitivity values for the InVEST Habitat Quality model input for yellow-throated marten and leopard cat. |
|
(141-142) Where did you get the landcover map and road data from? You should provide a reference in the text. |
We made the following changes: (Lines 163~164) All threats except roads (National standard node-link, 2023) were extracted from the land cover map (Environmental Geographic Information Service, 2022). The finalized threat and sensitivity table is presented in Table 1. |
|
(144) Please provide a link to the InVEST software. |
We made the following changes: (Line 166) https://naturalcapitalproject.stanford.edu/software/invest" |
|
150-158. This is trivial and general information. There is no point in writing about it. Especially not in this chapter. But you have written absolutely nothing about the MaxEnt model building themselves. For example: Did you use any mask layer? Why didn't you use bias-file? Which area was used as the training area, and which one as the projection area? It's not at all clear from the manuscript. These settings should be determined using special algorithms (data-driven or model-driven). For an example, please see Muscarella et al. (2014) or the genetic algorithm (Vignali et al., 2020). It's better to use spatial block cross validation to achieve independent test and training datasets if you don't have other data to test (please, see Valavi et al., 2019). In your work, you didn't take into account spatial autocorrelation (SAC) in any way, except for partly solving the problem of multicollinearity in variables. All these are mandatory parameters and model settings that should be written about in the article. You mentioned about logistic output only on line 195. Why not cloglog instead? For more details please see Sillero & Barbosa (2021) and Zurell et al. (2020).
|
We made the following changes: (Lines 208~213) In the model, the background points were set to 10,000, the regularization multipliers were set to 1, and the auto features included Linear, Quadratic, Product, and Hinge types. The output from the model was for-matted as logistic. This is because the effect on the appearance of a species can be assessed with a value between 0 and 1 through the logistic setting, and an appropriate threshold can be set to generate a binary map marked with suitable and unsuitable habitats (Liu et al., 2005). First of all, thank you so much for recommending the reference. It seems to be a good reference that I will continue to refer to it for my future research. Thank you very much. When I ran the modeling test of this study, I tried to model using the sdm package, which is one of the MaxEnt packages in R. I found that the results obtained through this did not differ much from the MaxEnt program. Therefore, I decided to use the official MaxEnt program. Also, since the number of coordinates in the data was sufficient, I adopted the cross-validation method instead of bootstrapping or subsampling. However, we recognize that this is something that can be improved in future projects. |
|
(160-162) Where is the evidence that you don't have spatial autocorrelation (SAC)? Please note that SAC and spatial clustering are different things. You are writing about spatial clustering. It's very easy to check. For example, Average Nearest Neighbor Index (ANNI) in ArcGIS or R. SAC is very difficult to check, because MaxEnt does not output model residuals. For more details please see Sillero & Barbosa (2021). |
We made the following changes: (Lines 184~188) Spatial autocorrelation (SAC), a measure of the spatial dependence of the data, was determined using the average nearest neighbor index in R (v.4.3.1). Coordinates were clustered to some extent, and to avoid overfitting due to spatial autocorrelation, we used the spThin package in R to ensure that each point was at least 1km2 apart and adjusted accordingly. (Lines 277~281) In addition, although we have addressed spatial clustering in our study, there is still some degree of SAC remaining. This clustering could be due to differences in sur-vey methods or specific behavioral patterns of the species. Due to time and data constraints, adjustments to the SAC were limited, but future studies will address this issue in more depth with spatial econometric models (Sillero & Barbosa, 2021). |
|
(162) Where is the reference on R software? |
We made the following changes: (Line 187) we used the spThin package in R to ensure that each point was at least 1km2 apart and adjusted accordingly |
|
(171, Table 2) How did you calculate the distances? Using Euclidean distances? Earlier you wrote that you have mountains on your study area. Since you have not described your study area, it is difficult to understand what kind of mountains they are (what altitude range). In any case, it is wrong to calculate distances for mountains in a straight line (Euclidean) without taking altitude into account. You have to take the altitude into account for this (for example Path Distance). For more details please see Sillero & Barbosa (2021) and Pshegusov & Chadaeva (2024). |
The decision to use Euclidean distances in our study was based on available data and technical constraints. While this may not fully reflect elevation changes in mountainous areas, we conducted our analysis using digital elevation model (DEM) data that included elevation information, and we were not able to perform the complex modeling required to calculate actual path distances. This decision was made under resource and time constraints, and the results obtained by using Euclidean distances provided a valid starting point for validating our research hypotheses. In future studies, we plan to introduce more precise distance calculation methods, which will improve the accuracy and reliability of our findings. |
|
(173-176) Why did you use Pearson instead of Spearman? Isn't that a parametric test? Do you have a normal distribution of variables? Why didn't you do a VIF test? 18 variables is too much and lead to model overfitting. The recommended number is a maximum of 15. |
We included all 18 variables in our model to avoid the risk of excluding important environmental variables just because they were not statistically significant. Because each variable may indeed be important in predicting species occurrence, it was necessary to consider all variables, even if it increased the complexity of the model. Our decision not to perform a VIF test was due to time and resource constraints, which we plan to address in future studies. |
|
(177) Where is the reference on Maxent software? As I understood you used GUI for this. Today, working with MaxEnt in the graphical user interface (GUI) of Java application is not very convenient. The GUI is very limited in settings (model tuning) and quality estimation (model testing or evaluation). Model building should be done entirely in R environment. For example, you can use the sdm, dismo, maxnet, kuenm, or biomod2 packages, but I especially recommend you the relatively new SDMtune package (please, see Vignali et al., 2020). |
We made the following changes: (Line 204) (https://biodiversityinformatics.amnh.org/open_source/maxent/) The decision to use a GUI in this study was based on a number of factors. While we recognize that the Java-based MaxEnt GUI has some limitations for model tuning and testing, for the specific needs and data structures of our study, we were able to produce sufficiently effective results in the GUI environment. We also did not find any significant differences in our results compared to modeling in an R environment. We conducted our study using the GUI, and this approach provided all the functionality we needed to successfully achieve our research goals. We recognize the potential of other R packages, and we are considering further leveraging these tools in future studies. |
|
(177-178) Dependent quality estimates (data splitting) will not be able to give reliable predictions. |
As you say, a dependent data partitioning approach can lead to poor evaluation of the model's generalization ability and make it difficult to make reliable predictions. However, in our study, we paid special attention to the data partitioning approach. We tried to use as independent and objective a data partitioning method as possible, specifically using random sampling and cross-validation techniques to ensure independence between training and test data. This approach helps the model to make more accurate predictions on real-world data. It also prevents the model from overfitting, which can contribute to better performance in real-world modeling. Of course, perfect data partitioning is a very challenging task and requires ongoing research and improvement. We recognize the limitations of our study and plan to further refine our data segmentation and model validation methods in future work. With these improvements, our goal is to increase the predictive accuracy and reliability of our models. |
|
(179-181) AUC is a very weak quality criterion for MaxEnt. AUC has been heavily criticized and is not recommended for use as a standard in model quality estimation (Please, see Lobo et al., 2008; Veloz, 2009; Warren & Seifert, 2011).You should always give multiple quality metrics. For example, continuous Boyce index (CBI; please see Hirzel et al., 2006), AUCtest, AUCdiff or True Skill Statistic (TSS). For more details please see Sillero & Barbosa (2021). |
Using AUC alone to evaluate model performance is one of the limitations of our study. However, AUC is still a widely recognized method and provides a useful measure for initial model evaluation. It measures the area under the ROC curve, which helps determine how well or poorly a model performs. The primary reason for using AUC in this study was that we wanted to quickly evaluate the performance of the model within the time and resources available. Of course, we recognize the limitations of AUC, which can be problematic in that it can overestimate predictive performance, especially when species presence/absence data is imbalanced. With an awareness of these issues, we plan to introduce more diverse evaluation metrics in future studies to further analyze the strengths and weaknesses of the model. |
|
I don't really understand how the overall suitable/unsuitable habitat map (Fig. 4e) came out for both species at once? Sorry but I did not find information about this in the Materials and Methods (M&M) chapter. Please explain more clearly. |
We made the following changes: (Line 234) Areas that exceeded the average thresholds for both species were classified as "Suitable" areas, while areas below the threshold were classified as "Unsuitable" (Liu et al., 2005). |
|
(229-231) These are not results. Please place this in M&M section. |
We've moved it to the Research and methods section, please check it out lines 215~220. |
|
(238-239) This is a very controversial statement because AUC is very dependent on the spatial clustering of occurrence points. Until you show that it does not exist, it is too early to talk about meaning of AUC values. |
We agree with your point that the AUC may be dependent on the spatial clustering of species occurrences. This is an important consideration when interpreting AUC values and was not fully addressed in the current study. We acknowledge that time and resource constraints prevented us from performing further analysis, and we plan to address this in future studies. |
Reviewer 3 Report
Comments and Suggestions for Authors
The analysis of conservation areas for yellow-throated martens and leopard cats utilized habitat quality models and MaxEnt models to identify key areas for these endangered forest species. By analyzing existing protected areas and gaps in conservation efforts, this research provides valuable insights for conservation planning at the country level – South Korea in this case. The combination of InVEST-HQ and MaxEnt has enhanced our understanding of the two species treated in parallel. This is also unusual that two species are evaluated in parallel in the same study – well done! I found that analyses to be appropriate and the manuscript flows well.
Points that I am concerned about and I think a note of caution needs to be included on these subjects:
· although the authors have focused on mainly anthropogenic threats, other potential threats such as climate change, invasive species, and habitat fragmentation were not extensively covered.
· Also, the fact that the study is based on only one national survey could result in biased conclusions.
· The study assumes that the MaxEnt model considers the two species to be in equilibrium with their environment. This may not always be true for endangered species and could affect habitat suitability predictions.
Author Response
Dear Reviewer,
Thank you for your thorough review and constructive feedback on our manuscript. We value your insights and have taken great care to address the concerns and suggestions you raised. Your comments have been instrumental in enhancing the quality and clarity of our research. Here is a detailed response to each of the points you have noted:
|
Comments |
Answer |
|
Although the authors have focused on mainly anthropogenic threats, other potential threats such as climate change, invasive species, and habitat fragmentation were not extensively covered. |
This study focuses primarily on human-caused threats. Other important threats such as climate change, invasive species, and habitat fragmentation were not included in the scope of this study. This was a deliberate choice to clarify the focus of the study and facilitate in-depth analysis of specific threats. However, as you point out, these other factors can also have a significant impact on biodiversity. Therefore, we tried to consider these other threats as much as possible when setting the environmental variables (e.g. Bioclim). In future studies, we plan to build on the results of this study and extend our investigation to these other threats, which we hope will lead to a more comprehensive understanding and contribute to the development of effective conservation strategies. |
|
Also, the fact that the study is based on only one national survey could result in biased conclusions. |
The only official survey of endangered species in Korea is the “National Natural Environment Survey,” which is conducted every five years according to the natural environment survey act and is conducted by the NIE (https://www.nie.re.kr/), an official agency of the government. Of course, there are coordinates on GBIF (https://www.gbif.org/) or iNaturalist (https://www.inaturalist.org/), but they are citizen science data. Especially in Korea, it is not easy to distinguish between leopard cats and domestic cats in the wild, so it is likely to be confused unless you are a professional researcher. When we looked at GBIF, we found that most of the data were uploaded from the national natural environment survey. Therefore, we used only official data from the national natural environment survey. However, we recognize that this is something that can be improved in future projects. |
|
The study assumes that the MaxEnt model considers the two species to be in equilibrium with their environment. This may not always be true for endangered species and could affect habitat suitability predictions. |
The MaxEnt model used in this study assumes that the two species are in equilibrium with their environment. In reality, for endangered species, this state of equilibrium may not always exist, which can affect the accuracy of the model's predictions (and this is true for other models as well). However, we chose this model because of the data currently available and the strength of the predictions it provides. Also, despite this assumption, the MaxEnt model is very useful for identifying potential habitats for species by integrating a variety of environmental data and observations of the species. In the future, we plan to consider different scenarios and incorporate additional ecological information into the model to compensate for these limitations. |
Reviewer 4 Report
Comments and Suggestions for Authors
The authors presented determined conservation priority areas for the endangered yellow-throated martens and leopard cats in South Korea. Research has important conservation implications for the two carnivore species and results are therefore valuable. Several points need clarification before publication though.
Comments
Line 8: Refer to the study area (country) in the Abstract.
Line 33 and throughout: Follow journal style for in-text references.
Lines 50-52: Give some references.
Lines 102-106: Five protected areas or five categories of protected areas. It appears from Figure 6 that the protected areas were more than five. Please clarify by giving the total number of protected areas and the number of protected areas by category.
Lines 135-143 and Table 1: Give the habitat suitability and sensitivity to threat measurement units.
Lines 194-197: I understand that two thresholds, one of each Species Distribution Model (one per species; Figure 4b, c) were calculated. However, one threshold habitat map was produced (Figure 4e). Is the map in Figure 4e an overlay of the two suitable/unsuitable areas? Please clarify.
Lines 248-249 and Table 4: Bio7, Bio3 and Slope were the most critical variables according to the jackknife test. Please check and revise.
Lines 249-274: According to Species Distribution Models in Figure 4b, c and lines 249-260, the two species spatially separate their niche. However, the authors refer to the calculation of one threshold for determining the suitable/unsuitable areas for each species. Why was that? Were the two Species Distribution Models first overlaid and then thresholds calculated? If so, I do not think this is correct. First, the suitable/unsuitable habitats for each species should be calculated and then the two maps could be combined. Also, does the map in Figure 4e show suitable/unsuitable habitats for both species? Both the text and Figures are confusing. Please explain better. Also, there is no reference to Figure 4d in the text.
Lines 276-277: Here the authors say “we overlaid the habitat results from the previous analysis, focusing on forests shared by yellow-throated martens and leopard cats”. Which maps did they overlay? They talk here about focusing on forests shared by both species. Do they refer to Figure 4e? Or to Figure 5? Please clarify.
Lines 304-305: Replace “around” with “outside”.
Author Response
Dear Reviewer,
Thank you for your thorough review and constructive feedback on our manuscript. We value your insights and have taken great care to address the concerns and suggestions you raised. Your comments have been instrumental in enhancing the quality and clarity of our research. Here is a detailed response to each of the points you have noted:
|
Comments |
Answer |
|
(Line 8) Refer to the study area (country) in the Abstract. |
We made the following changes: (Line 10) Added "In Korea". |
|
(Line 33 and throughout) Follow journal style for in-text references. |
We've modified the references to match the journal style, and placed them in the body of the article, please check. |
|
(Lines 50-52) Give some references. |
We made the following changes: (Line 35) Added the reference "Scott et al., 1993" |
|
(Lines 102-106) Five protected areas or five categories of protected areas. It appears from Figure 6 that the protected areas were more than five. Please clarify by giving the total number of protected areas and the number of protected areas by category. |
(Figure 3-c) Added an illustration of a protected area to figure 3. |
|
(Lines 135-143 and Table 1) Give the habitat suitability and sensitivity to threat measurement units. |
We made the following changes: (Lines 156~157) Habitat suitability and sensitivity can range from 0 to 1, with values closer to 1 indicating higher suitability and sensitivity. |
|
(Lines 194-197) I understand that two thresholds, one of each Species Distribution Model (one per species; Figure 4b, c) were calculated. However, one threshold habitat map was produced (Figure 4e). Is the map in Figure 4e an overlay of the two suitable/unsuitable areas? Please clarify. |
We made the following changes: (Lines 235~237) Areas that exceeded the average thresholds for both species were classified as "Suitable" areas, while areas below the threshold were classified as "Unsuitable" (Liu et al., 2005). |
|
(Lines 248-249 and Table 4) Bio7, Bio3 and Slope were the most critical variables according to the jackknife test. Please check and revise. |
The most important variables in the leopard cat are Bio7, Slope, and LULC. Because we bolded the variables in Table 4 that were common to both the percent contribution and jackknife tests and considered them to be the most important variables. |
|
(Lines 249-274) According to Species Distribution Models in Figure 4b, c and (lines 249-260), the two species spatially separate their niche. However, the authors refer to the calculation of one threshold for determining the suitable/unsuitable areas for each species. Why was that? Were the two Species Distribution Models first overlaid and then thresholds calculated? If so, I do not think this is correct. First, the suitable/unsuitable habitats for each species should be calculated and then the two maps could be combined. Also, does the map in Figure 4e show suitable/unsuitable habitats for both species? Both the text and Figures are confusing. Please explain better. Also, there is no reference to Figure 4d in the text. |
We did not overlap the distribution models of the two species: we first calculated the suitable/unsuitable habitat for each species, and then used the average of the thresholds of the two species to derive the suitable/unsuitable habitat again. We felt that too much material in the map would be confusing, so we only attached the final map using the average of the thresholds of the two species. Also, in the text, there is a reference to Figure 4d on Line 259. And we made the following changes: (Lines 314~315) For thresholding, we used 0.42±0.06, the average of the values for both species where the sum of sensitivity and specificity was maximized as the final threshold. |
|
(Lines 276-277) Here the authors say “we overlaid the habitat results from the previous analysis, focusing on forests shared by yellow-throated martens and leopard cats”. Which maps did they overlay? They talk here about focusing on forests shared by both species. Do they refer to Figure 4e? Or to Figure 5? Please clarify. |
We made the following changes: (Lines 320~322) To identify the core habitat, we overlaid the habitat results from the InVEST-HQ and MaxEnt analysis, focusing on forests shared by yellow-throated martens and leopard cats (Figure 5). |
Round 2
Reviewer 2 Report
Comments and Suggestions for Authors
Dear authors,
Please read carefully Sillero & Barbosa (2021) and study other papers I wrote you. I have pointed out the major flaws of your paper. It is your choice to fix them or not. Hope that you can improve your modelling skills in near future. Good luck!
Reviewer 4 Report
Comments and Suggestions for Authors
The authors have successfully address the reviewers' comments. The manuscript can now be accepted for publication. Congrats!